# Characterization of Celiac Disease-Related Epitopes and Gluten Fractions, and Identification of Associated Loci in Durum Wheat

**Francesca Taranto [1]**, **Nunzio D'Agostino [2]**, **Marcello Catellani [3,†]**, **Luca Laviano [3,‡]**, **Domenico Ronga [3,§]**, **Justyna Milc [3]**, **Barbara Prandi [4]**, **Fatma Boukid [4,‖]**, **Stefano Sforza [4]**, **Sara Graziano [5]**, **Mariolina Gullì [5]**, **Giovanna Visioli [5]**, **Nelson Marmiroli [5]**, **Franz-W. Badeck [6]**, **Anna Paola Minervini [7]**, **Ivano Pecorella [7]**, **Nicola Pecchioni [7]**, **Pasquale De Vita [7]** and **Enrico Francia [3,*]**

1   Institute of Biosciences and Bioresources (CNR-IBBR), Portici, 80055 Naples, Italy; francesca.taranto@ibbr.cnr.it
2   Department of Agricultural Sciences, University of Naples Federico II, Portici, 80055 Naples, Italy; nunzio.dagostino@unina.it
3   Department of Life Sciences, Centre BIOGEST-SITEIA, University of Modena and Reggio Emilia, 42122 Reggio Emilia, Italy; marcello.catellani@enea.it (M.C.); l.laviano@rijkzwaan.nl (L.L.); domenico.ronga@unimore.it (D.R.); justynaanna.milc@unimore.it (J.M.)
4   Department of Food and Drug, Centre SITEIA.PARMA, University of Parma, 43124 Parma, Italy; barbara.prandi@unipr.it (B.P.); fatma.boukid@unipr.it (F.B.); stefano.sforza@unipr.it (S.S.)
5   Department of Chemistry, Life Sciences and Environmental Sustainability, Centre SITEIA.PARMA, University of Parma, 43124 Parma, Italy; sara.graziano@unipr.it (S.G.); mariolina.gulli@unipr.it (M.G.); giovanna.visioli@unipr.it (G.V.); nelson.marmiroli@unipr.it (N.M.)
6   Genomics and Bioinformatics Research Centre, CREA-GB, 29017 Fiorenzuola d'Arda (Piacenza), Italy; franz-werner.badeck@crea.gov.it
7   Research Centre for Cereal and Industrial Crops, CREA-CI, 71122 Foggia, Italy; a.minervini15@studenti.uniba.it (A.P.M.); ivano.pecorella@crea.gov.it (I.P.); nicola.pecchioni@crea.gov.it (N.P.); pasquale.devita@crea.gov.it (P.D.V.)
*   Correspondence: enrico.francia@unimore.it; Tel.: +39-0522-522041
†   Present address: Department for Sustainability, CR ENEA-Trisaia, 75026 Rotondella (Matera), Italy.
‡   Present address: Rijk Zwaan B.V., 2678 KX De Lier (Westland), The Netherlands.
§   Present address: Centro Ricerche Produzioni Animali-CRPA S.p.A, 42124 Reggio Emilia, Italy.
‖   Present address: IRTA Institute of Agrifood Research and Technology, 17121 Monells (Girona), Spain.

**Abstract:** While durum wheat is a major food source in Mediterranean countries, storage (i.e., gluten) proteins are however responsible for celiac disease (CD), a serious autoimmune disease that occurs in genetically predisposed subjects. Different gluten epitopes—defined as "immunogenic" (IP) and "toxic" (TP) peptides—are involved in the pathology and their content in wheat grain depends on environmental and genetic factors. Detection of IP and TP is not trivial, and no work has been conducted so far to identify the genomic regions associated with their accumulation in wheat. In the present study, a genome-wide association study was performed on a durum wheat collection to identify marker–trait associations (MTAs) between 5730 high quality SNPs and the accumulation of CD-related peptides and gluten protein composition measured in two consecutive cropping seasons (2015/2016 and 2016/2017). High-molecular-weight glutenin subunits (HMW-GS) were more stable between the two years, and differences in total gluten proteins were mainly due to low-molecular-weight glutenin subunits (LMW-GS) and accumulation of gliadins. In the first instance, association tests were conducted on yellow pigment content (YP), a highly inheritable trait with a well-known genetic basis, and several significant MTAs were found corresponding to loci already known for being related to YP. These findings showed that MTAs found for the rest of the measured traits were reliable. In total,

28 significant MTAs were found for gluten composition, while 14 were found to be associated with IP and TP. Noteworthy, neither significant ($-\log10p > 4.7$) nor suggestive ($-\log10p > 3.3$) MTAs for the accumulation of CD-triggering epitopes were found on *Gli-A1/Glu-A3* and *Gli-B1/Glu-B3* loci, thus suggesting regulatory rather than structural gene effect. A PBF transcription factor on chromosome 5B, known to be involved in the regulation of the expression of CD-related peptides, was identified among the positional candidate genes in the LD-decay range around significant SNPs. Results obtained in the present study provide useful insights and resources for the long-term objective of selecting low-toxic durum wheat varieties while maintaining satisfactory gluten quality.

**Keywords:** durum wheat; association mapping; gluten protein; gliadin fraction; toxic peptides; immunogenic peptides

---

## 1. Introduction

Durum wheat (*Triticum turgidum* subsp. *durum* Desf.) world production is only 5–8% of the total wheat production, although it is an economically important crop and the species is primarily associated with pasta and semolina production. For this reason, durum wheat has been subjected to intense breeding activities. For much of the past century, the main goal of genetic improvement was the increase in grain yield, whereas only since the late 1970s breeders have directed their attention to improving grain quality traits [1]. The continuous release of new cultivars has improved end-use suitability of the species by keeping pace with the needs of the pasta industry and changes in consumers' preferences. In general, pasta quality is estimated from its color, cooking attributes, and sensory properties [2,3]. The cooking quality of pasta mostly depends on its protein content and on the quality of gluten proteins [4,5]. Storage proteins of grains, i.e., glutenins (Glu) and gliadins (Gli), encoded by multiple genes at complex loci on durum wheat chromosomes 1 and 6, are responsible for the strength and extensibility of the dough, respectively [6].

Glutenin subunits (GS) are large protein aggregates of either low molecular weight (LMW) or high molecular weight (HMW) polypeptides linked together by disulphide bonds [7]. Gliadins are alcohol-soluble proteins, which fall into four groups, ω, γ, and α/β, based on their electrophoretic mobility. Besides the total protein gluten content, HMW-GS/LMW-GS and gliadins/glutenins (Gli/Glu) ratios affect dough and pasta technological properties [6]. High amounts of HMW- and LMW-GS determine an increase in the mixograph dough strength, a condition to obtain pasta with increased cooked firmness and cooking stability [6]. Indeed, as reported by De Vita et al. [8], the durum wheat cultivars released in Italy during the last decades of the 20th century were selected for favorable allele combinations in Glu, such as the HMW-GS 7 + 8 allele encoded by the *Glu-B1* locus. Such a breeding goal has led to a significant improvement in the rheological performance of the semolina and therefore in the cooking quality of the pasta. However, in recent years, consumers raised doubts about the nutritional benefit of pasta produced with modern varieties. It has been hypothesized that the augmented incidence in gluten-related disease may have occurred because of indirect changes in wheat proteins caused by modern breeding [9], although conflicting evidence has been reported [10].

Several studies suggest that different gluten epitopes—derived from either total or partial digestion in celiac disease (CD) genetically predisposed individuals—are involved in the pathology: some are defined "toxic" as a consequence of their ability to induce damage to the intestinal mucosa [11], other peptides are known to be "immunodominant", i.e., they cause a strong reaction commonly in all patients [12]. So far, the genetic and physiological mechanisms of the disease are partly understood, and the only effective cure for celiac individuals is a life-long gluten-free diet [13]. In addition, thanks to sequence data as well as epitope-specific T cells and antibodies, the heterogeneity of epitope occurrence has been demonstrated at protein level and at species and plant variety level [14–16]. These studies indicated the presence of a large variation of CD toxicity among wheat species and cultivars, and fortunately,

this opens possibilities to produce non- or less toxic wheat varieties through traditional and new plant breeding techniques. Alternative strategies have been proposed to reduce wheat gluten CD-triggering properties. On the one hand, it could be pursued the design and selection of new varieties with satisfactory technological properties combined with limited (hopefully null) content of toxic and immunogenic peptides [17]. On the other hand, a series of good agronomic practices could be deployed to modify gluten composition and further lower the content of CD-triggering peptides. Recently, Ronga et al. [18] investigated environmental and genetic factors that affect the accumulation of wheat gluten peptides. Data analysis of a multi-environment trial of eight site-year combinations and six durum wheat cultivars revealed the strong effect of the environment and the influence of the genotype on the accumulation of toxic peptides. As far as immunogenic peptides are regarded, a marked variation was registered in the different sites with significant genotype-by-year and genotype-by-site interaction. At present, the organization of the Glu and Gli genes has been elucidated by using the genome sequence information available for the hexaploid and tetraploid wheat species [19–21]. Recent advances in structural and functional genomics have substantially improved the knowledge on gluten proteins, particularly on the role that *cis*- and *trans*-acting factors have in regulating gene expression in the grains [22,23]. The new high-throughput sequencing technologies led to the identification of a large number of molecular markers such as single nucleotide polymorphisms (SNPs) applicable in genome-wide association studies (GWAS) [24] with the aim to link specific genetic variations to phenotypes of interest and accelerate simultaneous improvement of wheat end-use and health-related traits. To date, GWAS were widely used to investigate the association with gluten protein composition in wheat [22]. As far as the authors know, so far no study has addressed the detection of loci associated with CD-triggering immunogenic and toxic epitopes obtained via simulated gastrointestinal digestion.

The present study aims at (i) identifying marker–trait associations (MTAs) for the accumulation of CD-related peptides, (ii) assessing the relationship between HMW-GS and LMW-GS composition, grain protein content, and gluten peptides, and (iii) identifying candidate genes related to the accumulation of toxic and immunogenic CD-related peptides with the long-term objective of developing useful molecular markers to select low-toxic durum wheat varieties.

## 2. Materials and Methods

### 2.1. Plant Materials and Experimental Field Trials

We studied 79 durum wheat cultivars, mainly representative of the Italian durum breeding programs from 1915 to 2010. However, it should be noticed that most of the genotypes (53 out 79) under investigation were released in the 1990–2010 period. Year of release (YOR), registered pedigree, and breeder for each cultivar were recorded and are available in Table S1.

Field experiments were carried out during two consecutive growing seasons (2015/2016 and 2016/2017) on a clay-loam soil (Typic Chromoxerert) at the experimental farm of CREA-CI Research Centre for Cereal and Industrial Crops, Foggia, Italy (41°27′44.9″ N 15°30′03.9″ E). For both years, the previous crop grown in the experimental field was durum wheat, and the seedling density was 350 seeds $m^2$. Fertilizer applications were made as in normal practice, at pre-sowing (36 kg N $ha^{-1}$; 92 kg P $ha^{-1}$ as ammonium bi-phosphate) and as top dressing (52 kg N $ha^{-1}$ as ammonium nitrate) at Zadoks growth stages (GS) 2.2 and 3.1, respectively [25]. For both growing seasons, weeds were controlled with the herbicides Tralcossidim (1.7 L $ha^{-1}$), Clopiralid + MCPA + Fluroxypyr (2.0–2.5 L $ha^{-1}$). The sowing dates for the two years were 16 December and 13 December in 2015 and 2016, respectively. Each entry was sown in plots of 2 m-long rows, spaced 50 cm, in a random complete block design with two replicates; arrangements were made to avoid all edge effects including shading by older, taller cultivars [26]. At the end of the two growing seasons, plants were manually harvested after the physiological maturity on 10 June 2016 and on 19 June 2017, respectively.

## 2.2. Weather Conditions

Maximum and minimum temperature, and precipitation were recorded on a daily basis at the meteorological station installed within 1.5 km from the experimental field (Figure 1). As heat accumulation and total rainfall from sowing to harvest in the two cropping seasons were markedly different, water availability for plants was characterized by simulated soil water content and by a Water Stress Index (WSI) [27] calculated as follows:

$$WSI = 1 - \min\ [1, (SWC - AWC)/0.7\ AWC] \tag{1}$$

With SWC, the actual soil water content, derived from a modelled soil water balance that takes into consideration the available water holding capacity (AWC) of the soil.

Graphic summary of agro-meteorological data and of agronomic and quality traits were obtained using the R graphics package ggplot2 [28], while the correlation matrix, based on Pearson's rank correlation coefficients between pairs of phenotypes, was produced using the corrplot package 0.84 in R.

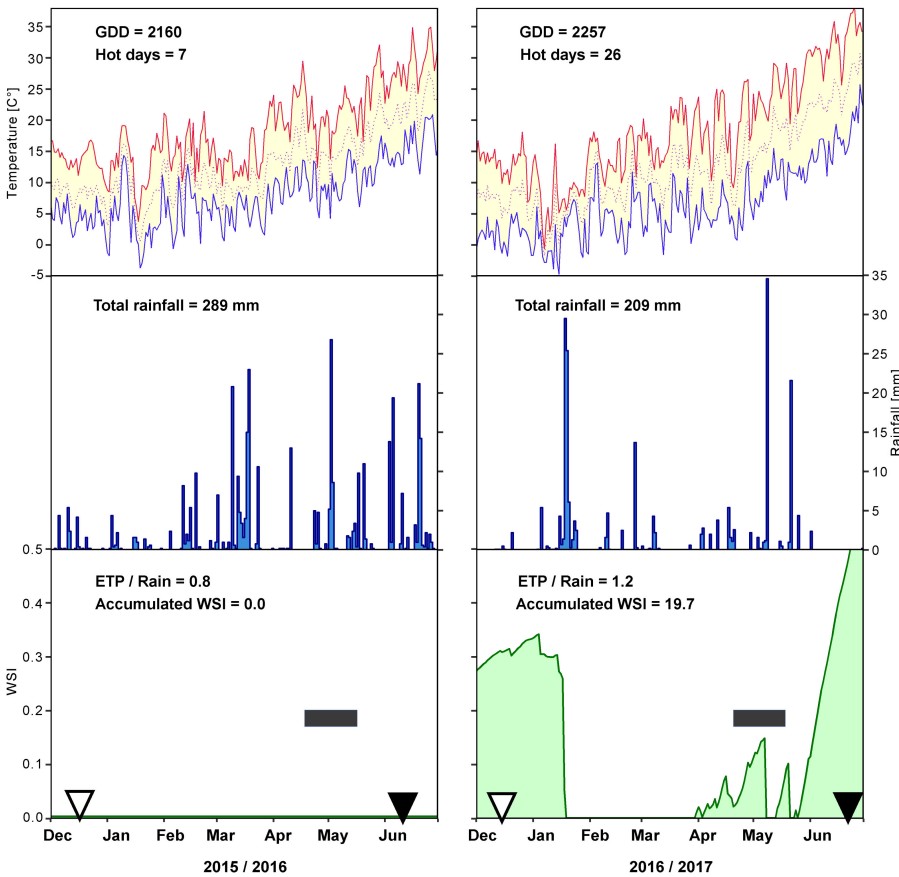

**Figure 1.** Agro-meteorological characterization of the 2015/2016 and 2016/2017 cropping seasons in Foggia, Italy. Maximum, minimum, and average temperature (°C, upper panels), rainfall distribution (mm, middle panels), and water stress index (WSI, lower panels) are reported. Empty and full arrowheads indicate sowing and harvest dates, respectively, while grey horizontal bars indicate the heading date period comprising all the 79 cultivars. GDD, growing-degree days; hot days, number of days in which $T_{max} \geq 28$ °C; ETP, potential evapotranspiration; all values are reported on a daily basis.

## 2.3. Grain Quality Analyses

An aliquot of the grains was ground on an experimental mill (Tecator Cyclotec 1093, Foss, Hillerød, Denmark) equipped with a 500 μm sieve and grain nitrogen content was determined following the standard Kjeldhal method. Percentage of protein content was calculated after multiplying Kjeldhal

nitrogen by 5.7 and was expressed on a dry weight basis by the AACC method 46-10. Yellow pigment (YP) (mainly carotenoids) content of the wheat was assayed using the water saturated n-butanol extracts with spectroscopic measurements at 435.8 nm by the AACC method 14-50.

The extractable gliadin, HMW-GS, and LWM-GS fractions were obtained by a sequential extraction procedure already described by Visioli et al. [29]. Three technical replicates were performed for each sample. In total, 30 mg of flour for each durum wheat sample was processed by adding 1.5 mL of propan-2-ol 55% (*v/v*) and mixing for 20 min at 65 °C. The supernatants were recovered following centrifugation for 5 min at 10,000 g and vacuum dried. The procedure was repeated two more times to remove possible gliadin residues. The pellets, containing the GS fraction, were resuspended in 400 μL of propan-2-ol 55% (*v/v*), 0.08 M Tris-HCl pH 8.3, and 1% DTT (*w/v*). After incubation at 60 °C for 30 min with continuous mixing, the supernatants, containing HMW-GS and LMW-GS fractions, were recovered by centrifugation for 5 min at 14,000 rpm. A proper volume of acetone was added to each sample to reach a final concentration of 40% (*v/v*), incubating for 10 min at room temperature. After centrifugation (5 min at 14,000 rpm) the supernatants, containing LMW-GS fraction, were transferred in a new tube and precipitated again with acetone (up to a final concentration of 80% (*v/v*)), whereas the pellets, containing HMW-GS fraction, were air-dried. Finally, all the extracted gliadins, HMW- and LMW-GS were dissolved in 50% (*v/v*) acetonitrile (ACN) with 0.1% (*v/v*) trifluoroacetic acid (TFA). Relative quantification was done in triplicate by the Bradford assay, using the iMark$^{TM}$ microplate reader (Bio-Rad, Boston, MA, USA).

*2.4. In Vitro Digestion of Durum Wheat Flours and Quantification of Gluten-Derived Immunogenic/Toxic Peptides*

Two technical replicates were used to quantify the immunogenic and toxic CD-triggering epitopes by a simulated gastrointestinal digestion of durum wheat flours performed following the consolidated method described by Minekus et al. [30]. Briefly, ≈1 g of flour was treated with 1 mL of simulated saliva (containing *α*-amylase from porcine pancreas type VI B at a final concentration of 75 U mL$^{-1}$, Merck KGaA (Darmstadt, Germany), incubating at 37 °C for 5 min. Afterwards, 2 mL of simulated gastric juice (pepsin from porcine gastric mucosa at a final concentration of 2000 U/mL, Merck KGaA (Darmstadt, Germany) were added to the mixture and incubated 2 h at 37 °C, preliminarily adjusting pH to 3.0. Lastly, 4 mL of duodenal juice (pancreatin from porcine pancreas at a final concentration of 100 U mL$^{-1}$, Merck KGaA (Darmstadt, Germany) and bile (10 mM from bovine and ovine, Merck KGaA (Darmstadt, Germany) were added, the pH value was corrected to 7.0, incubating the mixture at 37 °C for 2 h. After inactivation of enzymes (10 min at 95 °C), the digested samples were centrifuged at 3220× *g* for 45 min at 4 °C to recover the supernatant. The digested samples were again centrifuged (13,148 g for 10 min, 4 °C) before adding 5 μL of internal deuterated standard solution (TQQPQQPF(*d₅*)PQQPQQPF(*d₅*)PQ, 1.6 mM) to 295 μL of each sample supernatant. Reverse phase UPLC/ESI-MS analysis was carried out following the procedure described by Boukid et al. [31], quantifying the peptides by Stable Isotope Dilution Assay (SIDA). Ultrapure water was produced using an AlphaQ water purification system (Millipore, Burlington, MA, USA), and all the chemical reagents used were of recognized analytical grade. Toxic (TP) and immunogenic (IP) peptides were identified as in Prandi et al. [32]. IP that were identified contained the gliadin epitope DQ2.5-glia-γ5 (QQPFPQQPQ), while TP contained the amino acid sequences PSQQ, QQQP, QQPY, or QPYP. The total values for the two categories—referred to as TPT and IPT, respectively—plus two specific immunogenic peptides (IP2-SQQPQQPFPQPQ and IP3-QAFPQQPQQPFPQ) quantified separately were used for the downstream analyses.

*2.5. Genotyping*

For each sample, 50 mg of leaf tissue from each genotype was sampled, freeze-dried and shipped to TraitGenetics (Gatersleben, Germany) for DNA extraction using CTAB method and SNP Genotyping via the Illumina® iSelect 15 K Wheat platform, which contains 13,600 functional markers derived from the 90 K iSELECT SNP array [33]. Biallelic SNPs were selected and filtered for minor allele frequency (>1%) and call rate (>90%) using PLINK [34]. Markers were ordered according to the

physical map of Zavitan wild emmer [20] available at https://www.dropbox.com/sh/3dm05grokhl0nbv/AAC3wvlYmAher8fY0srX3gX9a?dl=0. Linkage disequilibrium-based pruning was performed to remove SNPs that were in linkage disequilibrium (LD; $r^2 < 0.8$) with one another using the SNP and Variation Suite (SVS) software package (version 8.4.0, Golden Helix Inc., Bozeman, MT, USA). The population structure was determined using ADMIXTURE version 1.23 [35] assuming K values from 1 to 10. Ten-fold cross-validation (CV) test and 1000 bootstrap replicates were run in order to determine the optimal K-value for population structure analyses. A membership coefficient threshold ($q_i > 0.55$) was used to assign individuals to a specific sub-population. Principal coordinate analysis (PCA) was then carried out using the SNP and Variation Suite (SVS) software package (version 8.4.0, Golden Helix Inc.) to visualize the genetic stratification of the durum wheat cultivars based on the sub-populations as detected by ADMIXTURE. A kinship matrix was calculated using the "Centered_IBS" method with default settings in TASSEL v. 5.0 [36]. LD was calculated between adjacent markers using the SNP and Variation Suite (SVS) software package (version 8.4.0, Golden Helix Inc.). The LD decay distance was determined across whole genome using a threshold of $r^2 = 0.20$ [37,38].

### 2.6. GWAS and Identification of Candidate Genes

SNP/traits association tests were performed for YP, gluten protein composition, immunogenic and toxic epitopes. Association tests were independently run on data of the two cropping seasons. For each genotype, the average of the replicates was considered. A compressed mixed linear model (CMLM) was run using the R package Genome Association and Prediction Integrated Tool (GAPIT) [39]. The model used the population structure as the fixed effect and a kinship (K) matrix as the random effect (Q + K). Phenotypic variation explained by associated SNPs was approximated by taking the difference between the likelihood ratio-based $R^2$ of the model with the SNP and the likelihood ration-based $R^2$ of the model without the SNP. Additionally, the Fixed and random model Circulating Probability Unification (FarmCPU) method was used to increase the statistical power [40]. To avoid false positives due to multiple testing, the MTAs were filtered following Bonferroni correction at $-\log 10p > 4.7$. In addition, MTAs with correction at $-\log 10p > 3.3$ were considered "suggestive MTAs". A graphical representation of significant MTAs on chromosomes was obtained using PhenoGram (http://visualization.ritchielab.org/phewas_views/plot).

The physical and genetic positions were assigned at associated loci on the basis of the emmer [20] and durum [21] genomes and on the tetraploid wheat consensus map [41], respectively. Significant MTAs were annotated by using the wild emmer high-confidence gene models. The size of SNP flanking regions was determined based on LD decay.

## 3. Results

### 3.1. Field Trials and Grain Quality Analysis

Rainfall during the crop growth cycle (i.e., from sowing to harvest) varied between the two growing seasons ranging from 289 mm in 2015/2016 to 209 mm in 2016/2017 (Figure 1). Potential evapotranspiration was lower than precipitation in 2015/2016, while it exceeded rainfall in the much drier 2016/2017. The first growing season was characterized by a more equally distributed precipitation, ensuring adequate water supply during the entire season from sowing to harvest with an accumulated WSI = 0. On the contrary, the growing season 2016/2017 was characterized by a lower amount of rainfall and two dry periods that however did not compromise the durum wheat growth cycle. A WSI per day value equal to 0.31 was estimated for the 35 days following sowing with an accumulated WSI = 10.89 in coincidence with the early plant development phases, that is crucial for crop establishment in Mediterranean environments. From the second half of January up to flowering time, the WSI dropped to 0, then rapidly increasing during the grain filling period (from the end of May to harvest) over 0.4. As for temperature, 7 and 26 hot days ($T_{max} \geq 28 \, °C$) were registered during the first and the second growing season, respectively (Figure 1). Although contrasting, the meteorological conditions of both

experimental years are not uncommon in Mediterranean environments, as revealed by long-term observations (since 1956, Figure S1).

The two cropping seasons influenced the variation of the measured phenotypic traits (Figure 2). In general, 2015/2016 was characterized by lower mean values (seven out of 12 traits) and by a narrower data distribution (10 out of 12 traits). The ample variation in Gli and LMW-GS in 2016/2017 influenced the derived traits (Gli+Glu, Gli/Glu, and HMW-GS/LMW-GS). Interestingly, a clear upward shift in the distribution of total immunogenic (IPT) and total toxic (TPT) epitopes (two- and 10-fold in terms of mean, respectively) was observed by comparing 2015/2016 with 2016/2017.

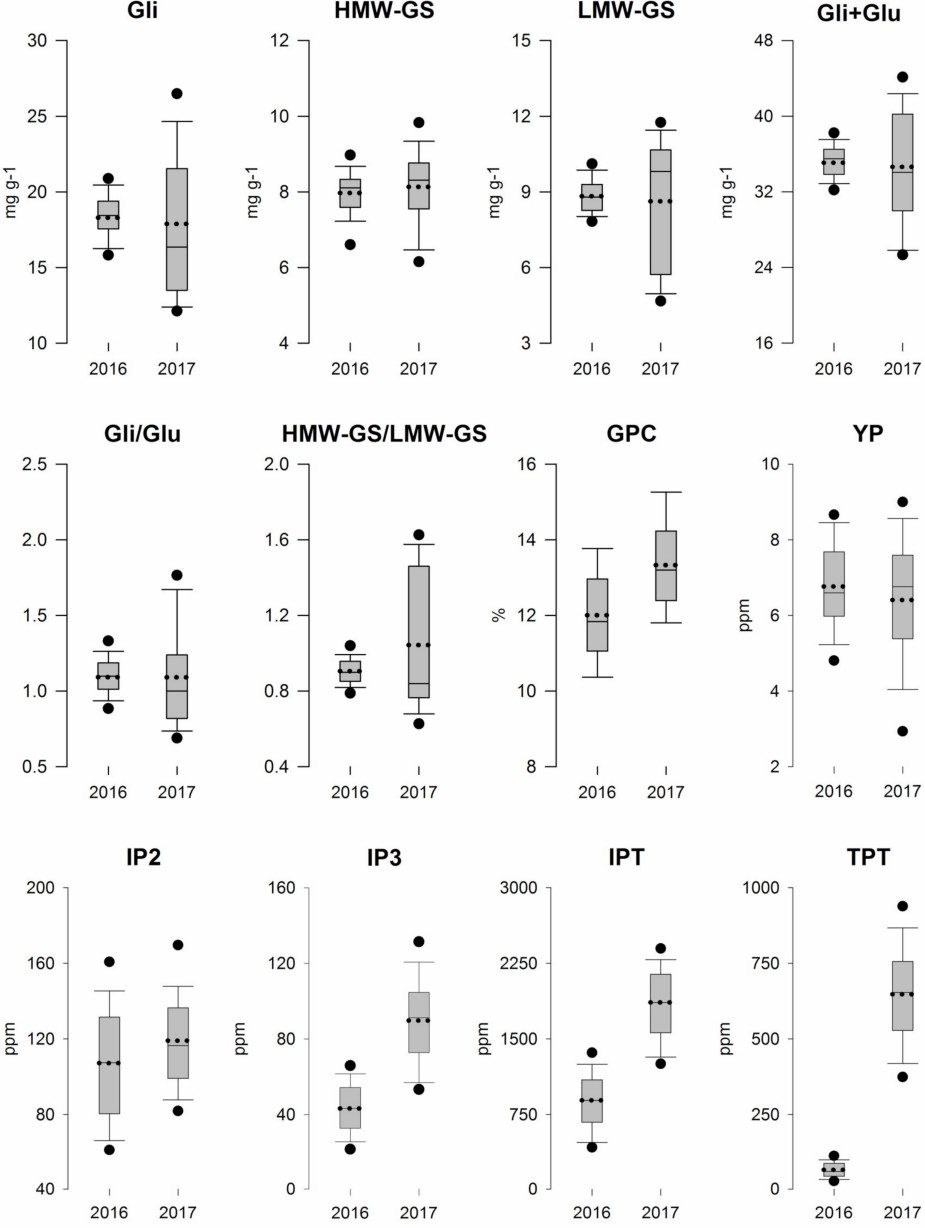

**Figure 2.** Box and whisker plots display variation in the two cropping seasons of gliadins (Gli; mg g$^{-1}$ flour), high and low molecular weight glutenins (either as fractions high-molecular-weight glutenin subunits (HMW-GS), low-molecular-weight glutenin subunits (LMW-GS), or total Glu; mg g$^{-1}$ flour), grain protein content (GPC; %), yellow pigment (YP; ppm), and peptides triggering the adaptive (IP2, IP3, IPT; ppm) and innate (TPT; ppm) immune response in celiac disease (CD) patients. Lower and upper dots indicate the 5th and 95th percentile, respectively, while the dotted line indicates the average value.

In the first cropping season, grain protein content was positively correlated with peptides triggering the adaptive (IP2, IP3, IPT) and innate (TPT) immune response in CD patients, while it was negatively correlated with Gli+Glu (Figure 3 and Table S4). YP was positively correlated ($r = 0.321$, $p = 0.035$) with the year of release (YOR) of the cultivars. Interestingly, YOR was negatively correlated with CD-related peptides ($r = -0.305$, $p = 0.038$ and $r = -0.210$, $p = 0.050$ for IPT.16 and TPT.16, respectively). Although the trend of correlation in 2016/2017 was not as clear as in the first season, YOR was still correlated with IPT.17 ($r = -0.274$, $p = 0.015$).

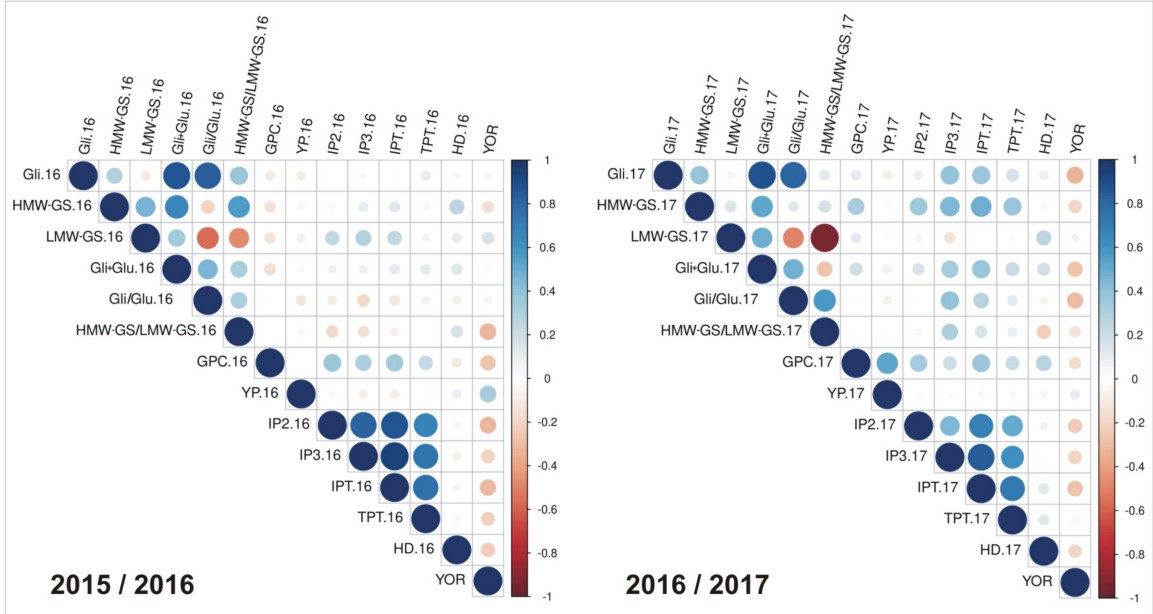

**Figure 3.** Pearson's rank correlation coefficients between pairs of traits measured in 2015/2016 and in 2016/2017 cropping seasons. Positive correlations are in blue and negative correlations in red. Color intensity (from light to dark) and the size of the circles (from small to big) are proportional to the correlation coefficients. GPC = grain protein content; Gli = gliadins; Glu = glutenins; HMW-GS = high molecular weight glutenin subunit; LMW-GS = low molecular weight glutenin subunit; YP = yellow pigment; IP = immunogenic peptide(s) and total (IPT); TPT = toxic peptide total; HD = heading date; YOR = year of release.

### 3.2. Population Structure and Linkage Disequilibrium

After filtering, 5730 polymorphic SNPs were employed for GWAS. LD-pruning removed 3101 SNPs, therefore only 2628 SNPs were used for population structure analysis by adopting the admixture model-based method with K ranging from 1 to 10. Cross-validation error indicated K = 4 ($CV_{error} = 0.758$) (Figure S2) as the most probable number of inferred sub-populations. The 19% of durum wheat cultivars were assigned to population P1, the 15.2% were assigned to populations P2 and P3, the 29.1% were assigned to population P4, and the remaining were tagged as admixed (Figure 4A). The clustering highly depended on the pedigree of durum wheat cultivars, grouping together cultivars characterized by different years of release but that shared the founders. Principal component analysis was used as an alternative way to investigate population stratification. The first two principal coordinates explained only 11% of genotypic variance (PC1: 5.85%, PC2: 5.15%) (Figure 4B). Combined analysis of genetic structure and PCA showed scattered position of genotypes across the axes, indicating that the cultivars belonging to P1 and P4 were genetically more different than those included in P2 and P3 (Figure 4B). A heatmap of the kinship matrix (K; individual relatedness) was visualized in Figure 4C. A pairwise comparison among the 5730 marker loci was computed to calculate the LD across the genome. The statistically significant threshold for $r^2 = 0.20$ was used to assess the rate of LD decay with physical distance. A genome-wide estimate of LD decay was ≈4 Mb (Figure S3).

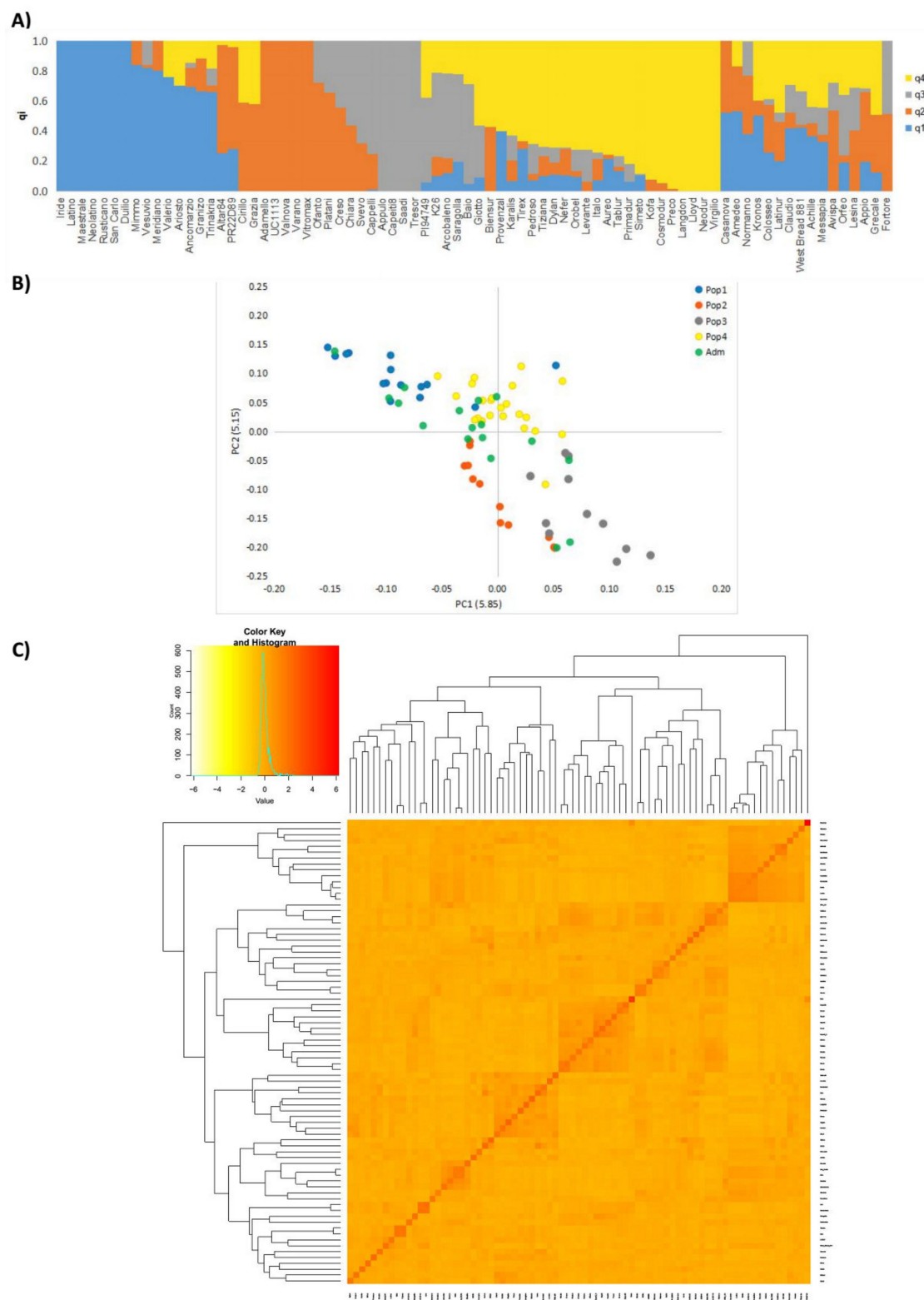

**Figure 4.** Population structure using 2628 single nucleotide polymorphism (SNP) markers. (**A**) Bar-plot describing the population structure at K = 4. Each accession is represented by a bar, which is partitioned into four colored segments whose length is proportional to the estimated membership coefficient ($q_i$). (**B**) Scatter plot of the top two principal coordinates (PC1 and PC2). Accessions are colored according to the sub-population they were assigned to by ADMIXTURE at K = 4. (**C**) Heatmap and dendrogram of a kinship matrix estimated using GAPIT. Colors show the distribution of values of relatedness in the whole kinship matrix.

### 3.3. Genome-Wide Association Studies

All the 5730 high quality SNPs were used for association tests between markers and traits (i.e., gluten protein composition, immunogenic and toxic epitopes). In order to verify the reliability of the genetic associations, due to the limited number of genotypes used in the present study, a preliminary marker–trait association analysis was conducted on yellow pigment content (YP), a highly inheritable trait with a well-known genetic basis [42–44]. Fifty MTAs are listed the Table S2. The CMLM method detected 17 MTAs for YP.16 ($-$log10$p$ > 5.422) and 25 for YP.17 ($-$log10$p$ > 4), out of which 17 were shared with YP.16. Significant regions were identified across the chromosomes 1A, 1B, 2A, 2B, 3B, 4B, 5A, 5B, 7A, and 7B (Table S2). Chromosomal regions frequently associated with YP in durum wheat [43,45,46] were confirmed in our work. The genetic associations detected in 2016 were stronger than in 2017. The marker Kukri_c7804_2504 was close to a gene that encodes for a farnesyl pyrophosphate synthase (FPPS) ($\approx$980 kb), which supplies precursors for the biosynthesis of essential isoprenoids like carotenoids [47]. The interval between the markers IAAV5683 and IACX5390 identified a region on chromosome 5A, where a QTL for semolina yellowness is annotated [43]. Finally, the marker wsnp_Ku_c11060_18147688 was located in a region on chromosome 7B where the major QTL for semolina yellowness and yellow pigment was detected by Patil et al. [48], Roncallo et al. [49], Fiedler et al. [50], and Colasuonno et al. [43].

As for genome-wide associations with the traits linked to the durum wheat gluten quantity and quality, 28 significant MTAs were found for gluten composition, whereas 14 were found to be associated with immunogenic and toxic epitopes (Table 1, Figure 5, Figures S4 and S5). According to the CLMLM model, three significant SNPs were detected for Gli.16, of which two were also detected in Gli+Glu.16. The two markers BobWhite_c19155_246 and wsnp_Ex_c4026_7281501 explain 26.9% and 32% of the phenotypic variations, respectively. The remaining MTAs were significant using the FarmCPU model. Six SNPs were detected for HMW-GS/LMW-GS.16 on chromosomes 6A and 6B, while 19 MTAs were found associated with HMW-GS/LMW-GS.17 on eight chromosomes (i.e., 1A, 1B, 2A, 2B, 3B, 4B, 5B, and 7B). Five SNPs were found associated with IP2.16 on chromosome 1B, while all were considered as suggestive MTAs for TPT.17 ($-$log10$p$ > 3.0). Seven MTAs were significantly associated with IP3.16 on chromosomes 1B, 2B, 4A, 6A, 6B, and 7B and only one with IP3.17 ($-$log10$p$ > 3.3) on chromosome 6B. Among these, two markers (BobWhite_c20073_382, chr. 1A and Excalibur_c92249_102, chr. 6B) were also detected with the total toxic epitopes (TPT.16).

Trans-acting factors are known to affect gluten gene expression. In our work, we identified some of these transcriptional factors mapping in chromosomal regions in LD with SNPs associated with traits related to protein components and epitopes. Several genes coding for ABC transporters involved in resistance to abiotic stress are on chromosomes 3B and 6A in LD with markers associated with HMW-GS/LMW-GS ratio 2015/2016 and 2016/2017. Moreover, different members of the cytochrome P450 superfamily, often involved in regulation of plant hormone homeostasis resulted in LD with markers on chromosomes 2A, 2B, 5B, and 6A (Kukri_c40953_658, RFL_Contig1385_326, IAAV5683, Tdurum_contig78006_158, RFL_Contig1385_341) associated with HMW-GS/LMW-GS.17. Several MTAs involving HMW-GS/LMW-GS.17 were in LD with regions on chromosomes 1A, 1B, 2A, 3B, and 5A, harboring genes coding for proteins involved in energy metabolism, starch metabolism, and photosynthesis (Table S5).

**Table 1.** Marker trait association for gluten component (Gli, HMW-GS, and LMW-GS) and CD-related peptide traits. SNP nomenclature, chromosome position, transcript ID, and annotation derive from the 'Zavitan' genome (WEWSeq v.1.0).

| Trait | SNP | Chr. | Position (bp) | cM | *p*-Value | $R^{2\,\S}$ | Allelic Effect | Transcript ID | Annotation |
|---|---|---|---|---|---|---|---|---|---|
| **Protein Component** | | | | | | | | | |
| Gli.16 | BobWhite_c19155_246 | 5A | 528,942,215 | 111.5 | $2.58 \times 10^{-7}$ | 0.269 | 2.478 | TRIDC5AG047350 | Protein STAY-GREEN, chloroplastic |
| | wsnp_Ex_c4026_7281501 | 5A | 529,160,493 | 111.5 | $2.58 \times 10^{-7}$ | 0.269 | −3.61 | TRIDC5AG047410 | Two-component response regulator-like PRR95 |
| | BS00003958_51 | 6B | 701,071,618 | n.a. | $1.69 \times 10^{-5}$ | 0.189 | 1.802 | TRIDC6BG073020 | Unknown function |
| HMW-GS/LMW-GS.16 [a] | Tdurum_contig42729_433 | 6A | 34,690,496 | 29.1 | $3.37 \times 10^{-5}$ | n.a. | 0.037 | TRIDC6AG008160 | MATE efflux family protein |
| | Tdurum_contig42729_380 | 6A | 34,690,549 | 29.1 | $3.37 \times 10^{-5}$ | n.a. | −0.037 | TRIDC6AG008160 | MATE efflux family protein |
| | Tdurum_contig78006_158 | 6A | 456,240,525 | 56.8 | $2.59 \times 10^{-5}$ | n.a. | 0.035 | TRIDC6AG037510 | Fatty acid oxidation complex subunit α |
| | Tdurum_contig76709_195 | 6A | 456,677,777 | 58.4 | $7.13 \times 10^{-5}$ | n.a. | 0.032 | TRIDC6AG037650 | α/β-hydrolases superfamily protein |
| | BS00036878_51 | 6A | 457,248,758 | 58.4 | $4.32 \times 10^{-5}$ | n.a. | 0.037 | TRIDC6AG037700 | Receptor-like kinase 1 |
| | IAAV7349 | 6B | 64,846,436 | n.a. | $3.37 \times 10^{-5}$ | n.a. | 0.037 | TRIDC6BG011970 | ARC6 |
| HMW-GS/LMW-GS.17 [a] | GENE-1214_288 | 1A | 349,690,319 | n.a. | 0.0000079 | n.a. | −3.308 | TRIDC1AG029170 | Armadillo repeat-containing protein 7 |
| | BS00065170_51 | 1A | 582,636,662 | 156 | 0.0000079 | n.a. | −3.944 | TRIDC1AG063120 | Plant protein of unknown function (DUF247) |
| | BS00035690_51 | 1B | 138,929,471 | 34.6 | 0.0000079 | n.a. | −3.083 | TRIDC1BG017610 | Glucan 1,3-β-glucosidase A |
| | Kukri_c40953_658 | 2A | 91,415,342 | n.a. | 0.0000079 | n.a. | −3.308 | TRIDC2AG017740 | Cytochrome P450 superfamily protein |
| | Kukri_c67546_279 | 2A | 91,421,659 | n.a. | 0.0000079 | n.a. | −3.308 | TRIDC2AG017760 | Sulfotransferase |
| | RFL_Contig1385_326 | 2B | 147,198,649 | n.a. | 0.0000079 | n.a. | −1.444 | TRIDC2BG021670 | RNA-binding protein 39 |
| | RAC875_c25375_236 | 3B | 140,912,116 | n.a. | 0.0000079 | n.a. | −2.281 | TRIDC3BG020330 | UDP-Glycosyltransferase superfamily protein |
| | Kukri_c7804_2504 | 3B | 467,024,111 | n.a. | 0.0000079 | n.a. | −2.281 | TRIDC3BG042750 | Chloride channel E |
| | IACX3426 | 3B | 698,180,277 | n.a. | 0.0000079 | n.a. | −2.281 | TRIDC3BG065770 | Undescribed protein |
| | Ku_c46571_2583 | 3B | 701,422,737 | 130.1 | 0.0000079 | n.a. | −2.281 | TRIDC3BG066320 | Dicer-like 1 |
| | Kukri_c13345_481 | 3B | 751,717,521 | 144.8 | 0.0000079 | n.a. | −2.281 | TRIDC3BG072410 | Mitochondrial substrate carrier family protein |
| | BS00060666_51 | 3B | 751,945,425 | 144.8 | 0.0000079 | n.a. | −2.281 | TRIDC3BG072450 | Homeobox protein LUMINIDEPENDENS |
| | BS00105878_51 | 3B | 762,885,605 | n.a. | 0.0000079 | n.a. | −2.281 | TRIDC3BG074270 | Unknown function |
| | CAP12_rep_c4571_181 | 4B | 11,639,988 | n.a. | 0.0000079 | n.a. | −1.586 | TRIDC4BG003010 | Undescribed protein |
| | Kukri_rep_c79943_189 | 4B | 541,798,790 | n.a. | 0.0000079 | n.a. | −1.586 | TRIDC4BG046050 | Undescribed protein |
| | IAAV5683 | 5B | 518,614,625 | n.a. | 0.0000079 | n.a. | −3.308 | TRIDC5BG052520 | Proteasome subunit α-type-7-B |
| | Excalibur_c3165_730 | 5B | 590,819,414 | 131.2 | 0.0000079 | n.a. | −3.308 | TRIDC5BG062980 | BTB/POZ domain-containing protein |
| | IACX5390 | 5B | 600,810,891 | 136.7 | 0.0000079 | n.a. | −3.308 | TRIDC5BG064250 | Actin depolymerizing factor 6 |
| | wsnp_Ku_c11060_18147688 | 7B | 119,625,169 | 57.7 | 0.0000079 | n.a. | −3.308 | TRIDC7BG015000 | RNA-binding protein 47 |
| Gli+Glu.16 | BobWhite_c19155_246 | 5A | 528,942,215 | 111.5 | $7.45 \times 10^{-6}$ | 0.32 | 4.875 | TRIDC5AG047350 | Protein STAY-GREEN, chloroplastic |
| | wsnp_Ex_c4026_7281501 | 5A | 529,160,493 | 111.5 | $7.45 \times 10^{-6}$ | 0.32 | −4.875 | TRIDC5AG047410 | Two-component response regulator-like PRR95 |

**Table 1.** *Cont.*

| Trait | SNP | Chr. | Position (bp) | cM | *p*-Value | $R^{2\,\S}$ | Allelic Effect | Transcript ID | Annotation |
|---|---|---|---|---|---|---|---|---|---|
| **CD-Related Peptides [a]** | | | | | | | | | |
| IP2.16 | BS00041355_51 * | 1B | 496,196,589 | n.a. | $1.25 \times 10^{-5}$ | n.a. | −1.669 | TRIDC1BG045920 | Unknown function |
| IP2.16 | wsnp_JD_c6331_7499060 * | 1B | 553,899,486 | n.a. | $6.36 \times 10^{-5}$ | n.a. | 8.735 | TRIDC1BG052690 | Undescribed protein |
| IP2.16 | Excalibur_rep_c66322_448 * | 1B | 553,901,557 | n.a. | $1.08 \times 10^{-5}$ | n.a. | 5.159 | TRIDC1BG052690 | Undescribed protein |
| IP2.16 | Excalibur_rep_c107047_605 * | 1B | 609,661,176 | 98.8 | $1.00 \times 10^{-4}$ | n.a. | 3.146 | TRIDC1BG060060 | Choline/ethanolamine kinase |
| IP2.16 | Tdurum_contig96049_200 * | 1B | 610,156,207 | 98.8 | $1.00 \times 10^{-4}$ | n.a. | 3.146 | TRIDC1BG060130 | Undescribed protein |
| IP2.16 | RAC875_c34012_983 * | 7B | 734,184,474 | n.a. | $6.09 \times 10^{-5}$ | n.a. | 2.568 | TRIDC7BG073100 | 1,4-α-glucan branching enzyme GlgB |
| IP3.16 | BobWhite_c20073_382 | 1B | 584,479,982 | 156.3 | $2.94 \times 10^{-14}$ | n.a. | 8.541 | Intergenic | |
| IP3.16 | RFL_Contig5290_1493 | 2B | 205,965,316 | 87.4 | $2.26 \times 10^{-11}$ | n.a. | 5.350 | TRIDC2BG028530 | HXXXD-type acyl-transferase family protein |
| IP3.16 | BobWhite_rep_c66361_594 | 4A | 715,175,771 | 167.5 | $5.67 \times 10^{-5}$ | n.a. | 2.875 | TRIDC4AG069440 | Protein CWC15 homolog |
| IP3.16 | BobWhite_c22086_444 | 6A | 8,529,327 | 4.3 | $2.69 \times 10^{-7}$ | n.a. | 3.921 | TRIDC6AG002330 | COPII coat assembly protein SEC16 |
| IP3.16 | RAC875_c17559_3102 ** | 6B | 120,052,990 | 54.4 | $2.35 \times 10^{-7}$ | n.a. | 2.858 | TRIDC6BG017260 | Endoglucanase 11 |
| IP3.16 | Excalibur_c92249_102 | 6B | 652,231,937 | 114.5 | $1.10 \times 10^{-7}$ | n.a. | 1.389 | TRIDC6BG062810 | Unknown function |
| IP3.16 | Kukri_c39759_102 | 7B | 542,938,246 | n.a. | $3.14 \times 10^{-6}$ | n.a. | 0.963 | TRIDC7BG048190 | Methionine-tRNA ligase |
| TPT.16 | BobWhite_c20073_382 | 1B | 584,479,982 | 156.3 | $6.50 \times 10^{-6}$ | n.a. | −5.119 | TRIDC1BG056400 | Auxin response factor 4 |
| TPT.16 | CAP8_rep_c8022_270 | 2A | 33,508,142 | 35.6 | $3.06 \times 10^{-6}$ | n.a. | 6.925 | TRIDC2AG007400 | Ribulose bisphosphate carboxylase small chain |
| TPT.16 | Excalibur_c92249_102 | 6B | 652,231,937 | 114.5 | $1.63 \times 10^{-5}$ | n.a. | −6.045 | TRIDC6BG062810 | Unknown function |

[§]: $R^2 = R^2$ of the model with the SNP—$R^2$ of the model without the SNP; [a]: FarmCPU model output does not include $R^2$ values. *: Suggestive MTAs for TPT ($-\log 10p > 3.0$) in 2016/2017; **: Suggestive MTA for IP3 ($-\log 10p > 3.3$) in 2016/2017.

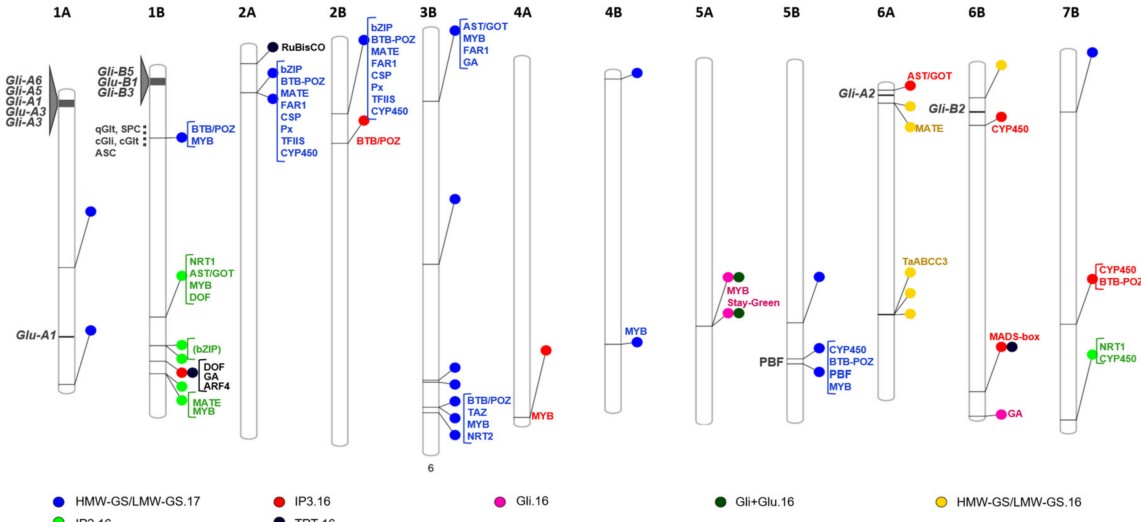

**Figure 5.** PhenoGram used to plot association results for gluten protein composition traits and immunogenic and toxic epitopes plot along 12 durum wheat chromosomes (chr. 3A and chr. 7A were omitted as no significant MTAs were found). Lines drawn on the chromosomes correspond to the physical location [20] of each MTA, and are connected to colored circles representing the phenotype(s) associated with the corresponding SNP. Genes/QTLs known in literature were reported on the left of each chromosome. Significant candidate genes were reported on the right. Gli = gliadins; Glu = glutenins; NRT1 = nitrate transporter; AST/GOT = aspartate aminotransferase; GAs = Gibberellin-regulated family protein; bZIP = basic leucine zipper; DOF = DNA binding with one finger; PBF = prolamin binding factor; ARF 4 = Auxin response factor 4; RuBisCO = ribulose-1 5-bisphosphate carboxylase oxygenase protein; CSP = cold shock protein, Px = peroxidase; CYP450 = Cytochrome P450 superfamily; NRT2 = high affinity nitrate transporter; TFIIS = Transcription elongation factor S-II; MYB = TaABCC3 = ABC transporter.

## 4. Discussion

Wheat gluten is composed of glutenin and gliadin proteins, which are recognised as the major grain storage proteins (GSPs), constituting about 60–85% of the total kernel proteins. Breeding success for high grain protein concentration (GPC) has been relatively hindered by its complex inheritance and large variation due to environmental effects. Moreover, wheat GPC is negatively associated with economic grain yield (GY) hampering attempts to improve these traits simultaneously. Selection for increased GY has probably countered gains in GPC during the past decades [51]. Studies based on the comparison of wheat cultivars released in different years (or breeding era) have shown that modern genotypes have reduced GPC compared with older ones [52,53]. It is thus important to develop wheat cultivars with well-balanced grain protein compositions to compensate for the low GPC of modern high-yielding accessions [22,54,55]. This trend is partially associated with the selection of favorable storage protein alleles, encoded at the six gliadin and glutenin loci, that started in the mid-1980s [56]. However, over the past two decades the role that specific toxic gluten protein fractions have had in triggering celiac disease and their perceived role in non-celiac gluten or wheat sensitivity has been proved [57].

In the present study, field experiments were carried out during two consecutive growing seasons (2015/2016 and 2016/2017). Grain quality analysis and in vitro digestion of durum wheat flours and quantification of gluten-derived immunogenic/toxic peptides were performed. Results of the first year, in no stress conditions, show correlation between IP2 content and year of release ($r = -0.317$, $p = 0.047$). Adjusting for heading date in partial correlation slightly improves the $r$-value ($r = -0.330$). These data follow the trend (more recent varieties, lower IP content) observed by Ficco et al. [10] for modern vs. ancient varieties. The influence of environment is less visible in no stress conditions as correlation ($r = -0.236$) between heading date and year of release were of borderline significance.

As in Mediterranean environments rainfall and water stress index are fundamental in ensuring the agronomic performance of winter cereal crops [58,59], changes in duration of developmental phases (i.e., vegetative, reproductive and grain filling) of durum wheat are among the well-known effects of breeding during the last century [60–62]. Favorable alleles at major genes regulating the length of the growing cycle (e.g., vernalization—VRN, photoperiod—PPD, and early maturity—EAM determinants) are advantageous for escaping terminal stress, and the best performing cultivars are generally early flowering [63]. In the growing season 2016/2017, higher water stress index was registered in coincidence with early developmental stages and from flowering until harvest. Correlation between IP2 and YOR showed lower values than in 2015/2016 suggesting that the effect was partially masked by environmental constraints. Moreover, adjusting for heading date in partial correlation slightly worsened the *r*-value ($r = -0.224$). This could mean that the environmental effect (WSI from flowering onwards) in the second year predominates over the genetic effect, and similar observation was recently obtained by Ronga et al. [18] in a multi-environment trial in Italy. Taking also into consideration the limited number of genotypes included in this study, we might hypothesize that this may have caused weaker significant marker–trait associations in 2016/2017 growing season.

Interestingly, by ranking the genotypes based on the accumulation of CD-related peptides across growing seasons separately, a group of varieties characterized by lower vs. higher values could be identified (Table S6). Combining this information with the GPC content, could pave the way for specific breeding programs aiming at varieties with lower CD-triggering epitopes without scarifying quality parameters.

It is well known that genotype dependent variation in GPC is by far lower than that induced during the grain filling (GF) period by environment and management practices [55]. Conditions such as $CO_2$ concentration, nitrogen availability, sunlight, temperature, and water availability determine GPC variation; however, the grain starch shows greater environment dependent variation, whereas the protein quantity remain relatively stable in wheat [64]. While grain starch synthesis depends on photosynthetic activity and thus is primarily influenced by the length of the GF period, the nitrogen required for the synthesis of proteins is mainly remobilized from the vegetative organs. Nitrogen remobilization and accumulation within the grain occurs predominantly in early GF stages. Stress conditions such as high temperature or drought induce premature senescence (shorter GF period) and thus lead to a decrease in the quantity of synthetized starch, which is accompanied by a higher grain nitrogen content [65]. Broad sense heritability ($H^2$) was calculated from linear regression (REML) analysis in which genotypes were fitted as random terms and the 'year' factor was used to describe the blocking of the trial. Similar $H^2$ values for YP and GPC (0.63 and 0.66, respectively) were in favor of the hypothesis that the protein quantity remains stable and the differences observed in percentage ratio are given by the decrease of grain starch quantity caused by the premature senescence induced by the environmental stress.

By using both protein composition and content of toxic and immunogenic CD-related peptides, a first tentative genome-wide association analysis was carried out using a collection of 79 durum wheat accessions. As far as the authors know, to date, no work has been conducted yet to identify loci associated with IP and TP accumulation in wheat. This may depend on the fact that the detection of toxic gluten epitopes is considered extremely complex [66]. We are aware that the primary limitation for GWAS was the restricted collection size and the long LD blocks that limited the resolution of association mapping [24,67]. A clear relationship between the effective sample size and the statistical power of any association study has been widely proved. Indeed, MTAs with a small effect may not be predicted. Clearly, the enlargement of the sample size improves the ability of predicting MTAs and is critical to the success of detecting causal genes. In our work, the phenotypic evaluation of IP and TP peptides represented the bottleneck, due to the expensive and laborious analysis required. To overcome the limit of the genome-wide association analysis and to restrict the false-positive results, we used the highly heritable trait YP as control. The association tests revealed several significant MTAs

identifying loci well-known for being related to YP [43,45,46]. These results encouraged us to believe that the associations found for components of gluten and celiac disease-related epitopes were robust.

The loci that control the expression of gliadins and glutenins are located on the chromosomes of groups 1 and 6 (Figure 5). In wheat the homoeologous composite loci *Gli-A1/Glu-A3* and *Gli-B1/Glu-B3* are located in a genomic region larger than 2 Mb on the short arms of group 1 chromosomes, and several genes coding for γ-gliadins (4–5), δ-gliadins (1–2), or ω-gliadins (3–8) are present in each Gli-1 region (for a recent review, see e.g., Wang et al. [23]). As expected, the short length of the epitopes studied in the present work and the complex organization of the loci do not allow tracing which specific genes determined their expression. However, by performing BLASTp searches of the IP2 and IP3 amino acid sequence against the *T. aestivum* (cv. 'Chinese Spring') and the *T. turgidum* (cv. 'Svevo') genomes (www.ensembl.org/), we retrieved γ-gliadin loci on chromosomes 1A and 1B. Noteworthy, we did not find neither significant nor suggestive MTAs on *Gli-A1/Glu-A3* and *Gli-B1/Glu-B3* loci when the accumulation of immunogenic and toxic CD-triggering epitopes was studied. This evidence reinforces the observation that in crops some QTLs represent the effects of regulatory rather than structural genes [22,68], and our MTA data may provide a further step in such direction.

The results of the association tests revealed that only one SNP marker (IAAV7349), related to HMW-GS/LMW-GS.16 and located on chromosome 6B, resulted in linkage (≈2.2 Mb) with α/β-gliadin genes (Table 1, Figure 5). We also confirmed several regions previously identified on 1B, 5B, 6A, and 6B chromosomes. On chromosome 1B, we found a significant association between BS00035690_51 and HMW-GS/LMW-GS.17. That SNP co-localized with the SSR markers *Xbarc187* previously associated with qGlt, SPC, cGli, cGlt, and ASC [22]. As expected, some MTAs were related to biosynthesis, trafficking, deposition, and secretion of gluten protein polymers. In particular these processes need the actions of some foldases such as peptidyl-prolyl *cis-trans* isomerases, PPIase (chr5B, HMW-GS/LMW-GS.17*), and of molecular chaperone (chr6A, HMW-GS/LMW-GS.16); during the export of protein aggregations, a specific coat protein complex named COPII (Chr 6A, IP3.16), located on the cytosolic face of the ER membrane, is required (Table S5) [69]. Other three regions associated with IP2.16, IP3.16, and TPT.16 were identified on chromosome 1B. The BobWhite_c20073_382 marker was associated with both IP3.16 and TPT.16 and it is located within the auxin response factor 4 gene. The region flanking the marker covers ≈1.6 Mb and included genes responsible for plant growth such as two genes involved in nitrogen metabolism (NRT1/PTR-5.10 and aspartate aminotransferase, AST/GOT) and gibberellin-regulated family protein (GA).

The accumulation of glutenins and gliadins is mainly controlled at the transcriptional level through a network of transcription factors. The promoter region of HMW-GS, LMW-GS, and α/β-gliadin genes contain several *cis*-elements, as the GCN4-like motif (GLM) and prolamin box (P-box), which are targeted by basic leucine zipper (bZIP) and DNA binding with one finger (DOF) transcription factors (TFs) [23,69–73]. Recent studies have provided substantial insights into the presence and function of conserved *cis*-regulatory modules (CCRM) in the promoters of HMW-GS genes [23,71,74]. Moreover, *trans*-acting factors that affect gluten gene expression have also been identified [22]; for example SPA and SHP (bZIP TFs) promote and repress the transcription of HMW-GS and LMW-GS genes, respectively [75,76]. Wheat *prolamin binding factor* (*WPBF*), encoding a DOF TF, is required for the efficient expression of LMW-GSs and gliadins in the grains [77–79]. A regulatory module MYB TF TaGAMyb and the histone acetyltransferase TaGCN5 regulated the expression of the HMW-GS gene *Glu-1Dy* by establishing a histone H3 acetylation pattern determining active gene transcription.

In our work, we identified some of these transcriptional regulators mapping in chromosomal regions in LD with SNPs associated with traits controlling storage protein components and CD-related peptides (Table S5). In particular, two genes encoding for DOF zinc finger proteins map on chromosome 1B in linkage with BS00041355_51 and BobWhite_c20073_382 markers associated with IP2.16 and TPT.16 traits. Two bZIP transcription factors were found on chromosomes 1B and 2A associated with IP2.16 and HMW-GS/LMW-GS.17, and several MYB transcription factors were found on chromosomes 1B, 2A, 3B, 4A, 5A, and 5B associated with gliadin, IP2, IP3, TPT content in 2015/2016 and with HMW-GS/LMW-GS

in 2016/2017. Interestingly, we also found that the two markers, namely Excalibur_c3165_730 and IACX5390 (chr. 5B), were in the region where the *Prolamin-box Binding Factor* (*PBF*) gene was mapped by Plessis et al. [22]. A missense mutation in DOF domain affects the level of gluten in barley and in wheat, with an important decrease [78]. Therefore, the PBF transcription factor could be considered a good candidate, as it is involved in the regulation of the expression of toxic and immunogenic CD-related peptides, to look at in order to select low-toxic durum wheat varieties.

Since environmentally induced changes in grain protein composition are associated with the altered expression of genes encoding Glu and Gli proteins [80], it could be presumed that terminal stress observed in 2016/2017 affected the expression of some regulatory proteins such as transcription factors (TF), heat shock (HSP) and late embryogenesis abundant (LEA) proteins that influence the expression of certain Gli genes causing, in turn, the increase of IPT and TPT after digestion. High throughput transcript sequencing of 61 durum wheat accessions showed quantitative and quantitative differences between the CD epitopes expressed in the endosperm; a few accessions showing a lower fraction of CD epitope-encoding α-gliadin transcripts [81]. Moreover, both the T-cell- and antibody-based assays of gluten protein fractions showed that differences in levels of expression occurred in the wheat accessions from diploid (AA, SS/BB, and DD genomes), tetraploid (AABB), and hexaploid (AABBDD) *Triticum* species [82].

The ratio HMW-GS/LMW-GS, a parameter influencing grain quality, resulted associated with many genes linked to abiotic stress response in plants. The proportion of LMW-GS showed large environmental variations in the two cultivation years (Figure 2) and this determined variation of the ratio. In particular, we found association of HMW-GS/LMW-GS.16 with two markers on chromosome 6A linked with MATE transporters that exert essential functions in nutrient homeostasis and ABA signaling under drought stress [83]. Several ABC transporters involved in resistance to abiotic stress are localized on chromosomes 3B and 6A in LD with markers associated with HMW-GS/LMW-GS in 2015/2016 and 2016/2017 such as TaABCC3.1 reported to mediate disposal of chlorophyll catabolites into the vacuole. This, in turn, would reduce chloroplast stress preventing cell damage by the photodynamic action of chlorophyll catabolites [84], and thereby support cell viability and resistance to such stress factors. This fits well with the observed positive effect of TaABCC3 transporters on wheat grain formation and the influence on head maturation, as evidenced in the VIGS (virus-induced gene silencing) studies and the gene expression data from Schreiber et al. [85].

Several players of the plant homeostasis network such as the cytochrome P450 (CYP450) superfamily members were in LD with markers on chromosomes 2A, 2B, 5B, and 6A (Kukri_c40953_658, RFL_Contig1385_326, IAAV5683, Tdurum_contig78006_158, RFL_Contig1385_341) associated with HMW-GS/LMW-GS.17. Among CYP450 superfamily functions, there is also the regulation of plant hormone homeostasis. Members of CYP450 family perform the key enzymatic steps in the biosynthesis of xanthophyll, the precursor of abscisic acid (ABA) [86]. A α/β-hydrolase (ABH) was associated with HMW-GS/LMW-GS.16 and with a marker on chromosome 6A (Table S5); ABH-fold serves as the core structure for phytohormone and ligand receptors in different pathways in plants [87]. In accordance with our thesis of the influence of premature senescence on GPC, several MTAs for HMW-GS/LMW-GS.17 were in LD with chloroplastic function on chromosomes 1A, 1B, 2A, 3B, and 5A, including proteins involved in energy metabolism, starch metabolism, and photosynthesis. Interestingly, on chromosome 5A the marker BobWhite_c19155_246 was associated with the Gli.16 and Gli+Glu.16, and it is on the chloroplastic protein STAY-GREEN (Table 1). Stay-green is a well-known heritable, delayed foliar senescence trait, which enables the plants to continue photosynthesizing also in stress conditions, and stay-green genotypes generally are more tolerant to abiotic stresses during GF stage, even if an extended maturity period may negatively affect the yield [88,89].

## 5. Conclusions

The insights and resources generated here provide useful information to (i) assess the relationship between grain protein content, gluten composition, and CD-related peptides; (ii) identify candidate

genes related to the accumulation of toxic and immunogenic CD-related epitopes, and (iii) develop useful molecular markers to select durum wheat varieties with reduced influence on the immunologic mechanisms of digestive disorders. This preliminary study will be used as a basis for larger GWAS in durum wheat, with a germplasm collection increased in old cultivars and landraces, improving the mapping resolution by expanding the number of SNP markers using a segregating population specially constituted to dissect the genetic basis of celiac disease gluten-related epitopes. In addition, as the traits under investigation are affected by environmental factors, it will become necessary to phenotype individuals in multiple environments in order to increase the reliability of SNP/trait associations. Overall, these actions should lead to increase the resolution, which is currently in LD-decay range, in order to bring out causal genes.

**Supplementary Materials:** The following are available online at http://www.mdpi.com/2073-4395/10/9/1231/s1, Figure S1: Frequency distribution of annual cumulative WSI and evapotranspiration to precipitation ratio, Figure S2: Cross-validation error estimates by ADMIXTURE, Figure S3: LD decay plot, Figure S4: Manhattan plot of gluten protein composition and immunogenic and toxic peptides, Figure S5: QQ plots of gluten protein composition and immunogenic and toxic peptides, Table S1: List of the durum wheat cultivars analyzed, Table S2: List of SNPs showing significant associations with YP, Table S3: List of SNPs showing significant associations with gluten protein composition and immunogenic and toxic peptides, Table S4: Pearson's rank correlation coefficients between pairs of traits in the two cropping seasons, Table S5: List of positional candidate genes identified within the LD-decay interval from the significant MTAs.

**Author Contributions:** Conceptualization, S.S., P.D.V. and E.F.; Data curation, M.C., L.L., F.B., A.P.M. and I.P.; Formal analysis, F.T., N.D., M.C., L.L., D.R., J.M., B.P., M.G., G.V., N.M. and F.-W.B.; Funding acquisition, S.S. and E.F.; Investigation, F.T., N.D., B.P., F.-W.B. and S.G.; Supervision, P.D.V.; Validation, F.T., N.D., B.P., F.-W.B. and S.G.; Writing—original draft, F.T., M.C. and E.F.; Writing—review and editing, F.T., N.D., M.C., L.L., D.R., J.M., B.P., F.-W.B., S.S., S.G., M.G., G.V., N.M., F.B., A.P.M., I.P., N.P., P.D.V. and E.F. All authors have read and agreed to the published version of the manuscript.

**Funding:** This research was financially supported by the Emilia Romagna region under the "POR FESR 2014–2020" program (Action 1.2.2. PG/2015/732409), project "Smart Wheat", and by the Ministry of Agricultural, Food and Forestry Policies (MIPAAF) with the project RGV-FAO.

**Acknowledgments:** The authors are grateful to Antonio Troccoli (CREA-CI) for providing weather data.

**Conflicts of Interest:** The authors declare no conflict of interest. The funders had no role in the design of the study; in the collection, analyses, or interpretation of data; in the writing of the manuscript, or in the decision to publish the results.

## Abbreviations

qGlt (total quantity of glutenin per grain), SPC (quantities of GSP, gliadin, or glutenin as percentages of Ntot, gliadin to glutenin ratio, and residuals to the relationships between the quantities of GSP, gliadin, or glutenin per grain and Ntot), cGli (quantities of each gliadin class as percentages of GSP or total gliadin), cGlt (cGlt, percentages of each glutenin subunit in GSP and HMW-GS to LMW-GS ratio), and ASC (allometric scaling coefficients of the relationships between the quantities of GSP, gliadin).

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
