# Peer review of "Characterization of Celiac Disease-Related Epitopes and Gluten Fractions, and Identification of Associated Loci in Durum Wheat"

_agronomy, doi:10.3390/agronomy10091231_

Round 1

Reviewer 1 Report

The objective of identifying genes related to the accumulation of toxic and immunogenic CD-related peptide in durum wheat is really an attractive topic since almost no such researches were reported. Authors also did pretty well for introducing the background information, however, it's really disappointing to see the experimental design and methods used in this research were poor and the results were not very reliable across datasets from two grown seasons. Here are few critical issues for this research:

1) very limited number (79) of genotypes used for GWAS. Even though authors indicated the GWAS on yellow pigment content, a highly inheritable trait as indicated by authors, detected 25 loci with some common loci in data from both grown seasons, and few were coincided with reported QTLs,  this only supports small population could be good for traits have high heritability or qualitative traits. But GWAS for traits with low heritability or low variations, a panel containing a good size of genotypes is required. It thus not surprising to see the GWAS in this research have so many loci were significant for Gli, HMW-GS and LMW-GS, and others, but they were not consistent in data from two seasons;

2) This research actually conducted GWAS for total quantity of raw amount of Gli, HMW-GS, LMW-GS and others in 79 durum wheat varieties. Data were collected from two seasons but no number of replicates were mentioned. No any statistical analysis were conducted for checking data quality. No any quality control were mentioned regarding phenotype accuracy; 

3) results interpretation is poor. Authors indicated "two markers BobWhite_c19155_246 and wsnp_Ex_c4026_7281501 explain 26.9% and 32% of the phenotypic variation", but actually this is the variation explained by the model not the marker. Table 1 listed all markers significant but with R2 and Allelic effect mostly showed as "n.a.",  indicated the problems in data analysis and results interpretation

Author Response

REVIEWER #1

The objective of identifying genes related to the accumulation of toxic and immunogenic CD-related peptide in durum wheat is really an attractive topic since almost no such researches were reported. Authors also did pretty well for introducing the background information, however, it's really disappointing to see the experimental design and methods used in this research were poor and the results were not very reliable across datasets from two grown seasons. Here are few critical issues for this research:

Reply: We appreciated the time the reviewer has spent in reading and revising this manuscript and we are grateful to the reviewer for their valuable comments. We believe that the received suggestions helped a lot to improve the overall quality of our manuscript. Furthermore, we have gone through all concerns, tried to clarify each point, and modified the title (which in our opinion is now better tailored on the content of the manuscript). All changes in the revised version of the manuscript are kept visible in the text.

1) very limited number (79) of genotypes used for GWAS. Even though authors indicated the GWAS on yellow pigment content, a highly inheritable trait as indicated by authors, detected 25 loci with some common loci in data from both grown seasons, and few were coincided with reported QTLs,  this only supports small population could be good for traits have high heritability or qualitative traits. But GWAS for traits with low heritability or low variations, a panel containing a good size of genotypes is required. It thus not surprising to see the GWAS in this research have so many loci were significant for Gli, HMW-GS and LMW-GS, and others, but they were not consistent in data from two seasons;

Reply: We are well aware that the number of genotypes used for this preliminary study is quite limited. The accessions studied in this paper represent a sub-sample of a larger collection of durum wheat genotypes already characterised by some of the Authors in previous studies: (Laidò et al. (2013) Genetic diversity and population structure of tetraploid wheats (Triticum turgidum L.) estimated by SSR, DArT and pedigree data. PLoS One 8:e67280; Taranto et al., (2020) Whole Genome Scan Reveals Molecular Signatures of Divergence and Selection Related to Important Traits in Durum Wheat Germplasm. Front. Genet. 11:217). The whole collection included indigenous Italian landraces, old cultivars, and modern cultivars. In the present study, we mainly focused our attention on modern varieties to verify that sufficient variation in CD-related epitopes exists and could be harnessed in future breeding approaches. Moreover, as the two growing seasons have been very different (likewise two contrasting sites), we obtained interesting indications for further researches. In particular, we are confident that the multiple coincidences between significant and suggestive MTAs could be validated in a more comprehensive study that is being currently undertaken. A suitable sentence has been added to the conclusion section (L541-543). Finally, regulatory genes rather that Gli/Glu genes are our long-term target, and as there is a lot of interest for the topic, we have preferred to present our results now instead of waiting for more information.

2) This research actually conducted GWAS for total quantity of raw amount of Gli, HMW-GS, LMW-GS and others in 79 durum wheat varieties. Data were collected from two seasons but no number of replicates were mentioned. No any statistical analysis were conducted for checking data quality. No any quality control were mentioned regarding phenotype accuracy;

Reply: We thank the reviewer for giving us the opportunity to better clarify the information given in the paper. Multiple replicates for checking data quality were used both for the analysis of gluten protein fractions and for peptidomic analysis. Both paragraph for gluten fractions and gluten peptides have been revised:

- Regarding gluten fractions (L151-172) the paragraph has been slightly rewritten to improve readability. As at L158 and according to Visioli et al. (2016), “three technical replicates were performed for each sample”. Some of the co-authors in other publications have already described the same protocol. For example: Visioli et al. (2018) Journal of the Science of Food and Agriculture, 98 (6), 2360-2369; Graziano et al. (2019) Technological quality and nutritional value of two durum wheat varieties depend on both genetic and environmental factors. Journal of agricultural and food chemistry, 67.8: 2384-2395.

- As for the immunogenic and toxic peptides (L173-195), the digestions and relative quantifications of the gluten peptides have been made in duplicate in accordance with an experimental protocol, which we have developed previously and which we now consider consolidated. See, for example, some publications that we obtained on the subject (Prandi et al. (2019) J Am Soc Mass Spectrom 30(8):1481-1490; Ronga et al. (2020) Eur J of Agron 118, August 2020, 126091).

results interpretation is poor. Authors indicated "two markers BobWhite_c19155_246 and wsnp_Ex_c4026_7281501 explain 26.9% and 32% of the phenotypic variation", but actually this is the variation explained by the model not the marker.

Reply: We thank the reviewer for giving us the opportunity to better clarify this point. Indeed we accidentally omitted the information that the two values of R2 (26.9 and 32) derive from the difference between the likelihood ratio-based R2 of the model with the SNP and the likelihood ration-based R2 of the model without the SNP. A suitable sentenced has been added to the manuscript.

Table 1 listed all markers significant but with R2 and Allelic effect mostly showed as "n.a.",  indicated the problems in data analysis and results interpretation.

Reply:  R2 values are reported only for MTAs outputted by the CMLM analysis, whereas they are missing in the case of MTAs obtained by applying FarmPCU, as association tests in FarmCPU are based on a fixed linear model. That is why R2 values were not automatically provided by FarmCPU (Please, see here: https://journals.plos.org/plosgenetics/article?id=10.1371/journal.pgen.1005767). On the other hand, FarmPCU returns allelic effect for each SNP and we did include this missing information in Table 1.

Reviewer 2 Report

The manuscript by Taranto et al. is a rather interesting piece of scientific work, proposing markers in durum wheat that can potentially be used to breed varieties less aggressive for celiac disease patients.

The main weakness of their work is the small number of markers and varieties used for a GWAS study. They do admit this and used the elegant strategy to mitigate it: detecting markers underlying the Yellow Pigment content trait and see if they match well-established markers or chromosome regions for this trait. This, of course, does not automatically guarantee that the markers detected are indeed associated with the other traits tested, but it’s a good start and the scientific community can later validate the markers found and conduct follow-up studies.

It would have been interesting to test durum landraces. Some of the markers used may have been genotyped in a series of landraces (35K SNP array) and are publicly available , so maybe it would be easy to do a follow-up study quantifying the same traits in these genotyped landraces.

Some suggestions: in the Methods section it is not completely clear if the phenotype measurements used for GWAS were from one of the two sowing years, an average of these two years or if two different GWAS were produced, one for each year. Based on line 299 I assume the latter but please make this clearer.

Also, DNA for genotyping was extracted how? From leaves? Which extraction method was used? Which tissue (leaf, hypocotyl, seed)? And was DNA extracted from plants in the first or second season? Please clarify this.

I’m also a bit concerned that so many genes are presented in Table S5 that one can look for which ones fit a narrative better while ignoring other genes that are possibly in LD with the markers that are not so “convenient”. While this may be inevitable, it would be good to see the authors proposing a future methodology to improve the resolution of these associations or to elucidate the genetic basis of these traits (even if it’s just tentative, of course).

Overall a commendable effort that merits publication in Agronomy.

Author Response

REVIEWER #2

The manuscript by Taranto et al. is a rather interesting piece of scientific work, proposing markers in durum wheat that can potentially be used to breed varieties less aggressive for celiac disease patients.

The main weakness of their work is the small number of markers and varieties used for a GWAS study. They do admit this and used the elegant strategy to mitigate it: detecting markers underlying the Yellow Pigment content trait and see if they match well-established markers or chromosome regions for this trait. This, of course, does not automatically guarantee that the markers detected are indeed associated with the other traits tested, but it’s a good start and the scientific community can later validate the markers found and conduct follow-up studies.

Reply: We thank the reviewer for the valuable suggestions and comments, which we believe they helped a lot to improve the overall quality of our manuscript. Furthermore, we have gone through all concerns, tried to clarify each point, and modified the title (which in our opinion is now better tailored on the content of the manuscript). All changes in the revised version of the manuscript are kept visible in the text.

It would have been interesting to test durum landraces. Some of the markers used may have been genotyped in a series of landraces (35K SNP array) and are publicly available , so maybe it would be easy to do a follow-up study quantifying the same traits in these genotyped landraces.

Reply: We thank the reviewer for the suggestion. We are aware that this is a preliminary study and that a follow-up should bring additional value to this research topic. The accessions studied in this paper represent a sub-sample of a larger collection of durum wheat genotypes already characterised by some of the Authors in previous studies: (Laidò et al. (2013) Genetic diversity and population structure of tetraploid wheats (Triticum turgidum L.) estimated by SSR, DArT and pedigree data. PLoS One 8:e67280; Taranto et al., (2020) Whole Genome Scan Reveals Molecular Signatures of Divergence and Selection Related to Important Traits in Durum Wheat Germplasm. Front. Genet. 11:217). The whole collection included indigenous Italian landraces, old cultivars, and modern cultivars. In the present study, we mainly focused our attention on modern varieties to verify that sufficient variation in CD-related epitopes exists and could be harnessed in future breeding approaches. For this reason, we have included a few lines on the possibility of exploiting the underexplored genetic variability of landraces in future GWAS (L541-L543). Moreover, due to the multiple coincidences between significant and suggestive MTAs revealed in the present study, a more comprehensive study that include also a series of landraces is being currently undertaken.

Some suggestions: in the Methods section it is not completely clear if the phenotype measurements used for GWAS were from one of the two sowing years, an average of these two years or if two different GWAS were produced, one for each year. Based on line 299 I assume the latter but please make this clearer.

Reply: As grasped by the reviewer, association tests were independently run on data of the two growing seasons. Since this was not sufficiently understandable, we made it clearer in the Methods section (L218).

Also, DNA for genotyping was extracted how? From leaves? Which extraction method was used? Which tissue (leaf, hypocotyl, seed)? And was DNA extracted from plants in the first or second season? Please clarify this.

Reply: We are very sorry for having missed those details. Missing information has been included in the text (L197). As the species is autogamous and we studied a collection of varieties, DNA was extracted the first year.

I’m also a bit concerned that so many genes are presented in Table S5 that one can look for which ones fit a narrative better while ignoring other genes that are possibly in LD with the markers that are not so “convenient”. While this may be inevitable, it would be good to see the authors proposing a future methodology to improve the resolution of these associations or to elucidate the genetic basis of these traits (even if it’s just tentative, of course).

Reply: We add a comment in conclusion section (See lines 548-551).

Overall a commendable effort that merits publication in Agronomy.

Reply: We thank once more the reviewer for their positive feedback.

Reviewer 3 Report

The manuscript presents original data on the mapping of loci associated with proteins responsible for the development of celiac disease. The scientific research is original. To date, no such studies have been carried out on varieties of durum and bread wheat. The introduction clearly presents the state of research. The Materials and Methods section describes in detail all the approaches and statistical programs used for the assessment of phenotypic traits and bioinformatic evaluation of the results. The reliability of the results is beyond doubt, since the authors used all the necessary criteria to search for MTAs.
A few minor notes on the manuscript. A genome-wide association study was carried out on the basis of an assessment of the traits over two seasons; however, it is not clear from the manuscript whether all associations were identified in both years. If not, it is advisable to make a check in Table 1.
The authors suggest that the obtained results allow to select wheat varieties with a low level of toxic peptides and a satisfactory gluten quality. It would be interesting for researchers to know which varieties from Table S1 have these characteristics.
From the statistical point of view, the population size is not enough to identify reliable MTAs for such a complex trait as gluten intolerance. In conclusion, the authors indicated that this work is preliminary and obtained results should be re-tested using the increased set of wheat varieties. However, I recommend adding a small section in the discussion chapter concerning the sample size and its effect on statistical significance.

Author Response

REVIEWER #3

The manuscript presents original data on the mapping of loci associated with proteins responsible for the development of celiac disease. The scientific research is original. To date, no such studies have been carried out on varieties of durum and bread wheat. The introduction clearly presents the state of research. The Materials and Methods section describes in detail all the approaches and statistical programs used for the assessment of phenotypic traits and bioinformatic evaluation of the results. The reliability of the results is beyond doubt, since the authors used all the necessary criteria to search for MTAs.

Reply: We thank the reviewer for appreciating our work as well as for their positive comments and valuable suggestions. Please, find below our responses to each of your minor comments/notes. We have modified the title, which in our opinion is now better tailored on the content of the manuscript. The text has been modified according to the comments/suggestions by the Editor and Reviewers, and all revisions are kept visible in the text.

A few minor notes on the manuscript. A genome-wide association study was carried out on the basis of an assessment of the traits over two seasons; however, it is not clear from the manuscript whether all associations were identified in both years. If not, it is advisable to make a check in Table 1.

Reply: As grasped by the reviewer, association tests were independently run on data of the two growing seasons. Since this was not sufficiently understandable, we made it clearer in the text (L218).

The authors suggest that the obtained results allow to select wheat varieties with a low level of toxic peptides and a satisfactory gluten quality. It would be interesting for researchers to know which varieties from Table S1 have these characteristics.

Reply: We thank the reviewer for pointing out this shortcoming. One way ANOVA and on Duncan's multiple range test have been per performed to rank the accumulation of immunogenic and toxic peptides. Results have been summarized in the new Table S6 that has been added to the Supplementary Tables file. A suitable sentenced has been added to the manuscript (L401-404).

From the statistical point of view, the population size is not enough to identify reliable MTAs for such a complex trait as gluten intolerance. In conclusion, the authors indicated that this work is preliminary and obtained results should be re-tested using the increased set of wheat varieties. However, I recommend adding a small section in the discussion chapter concerning the sample size and its effect on statistical significance.

Reply: Following reviewer’s suggestion, we did include a few lines in the discussion section (L426-430).

Reviewer 4 Report

This manuscript presents an interesting study on the celiac disease epitopes in durum wheat. In general, the manuscript is well write and the data showed are easily understood. However, I think that the figure 3 (p.8 ) could be changed by tables, because the size and colour are difficult to see by the most of readers.

Author Response

REVIEWER #4

This manuscript presents an interesting study on the celiac disease epitopes in durum wheat. In general, the manuscript is well write and the data showed are easily understood. However, I think that the figure 3 (p.8 ) could be changed by tables, because the size and colour are difficult to see by the most of readers.

Reply: We thank the reviewer for appreciating our work. We have modified the title, which in our opinion is now better tailored on the content of the manuscript.

Although we understand the criticism made on figure 3, we believe we should keep the figure because it offers an immediate overview of the correlations between the measured traits. On the other hand, the data the figure is based on, is presented in the supplementary Table S4 where for each pair-wise comparison the correlation value and the corresponding p-value are reported.

Round 2

Reviewer 1 Report

The revision doesn't improve the manuscript much. The new title is even more confusing, "identification of associated loci in durum wheat" doesn't give any clear meaning.

Overall, not too much useful information can be obtained from this research and no clear conclusion can be confidently made. Authors draw following conclusions:

i) assess the relationship between grain protein content, gluten composition, and CD-related peptides; (what kind of relationship was concluded?)

ii) identify candidate genes related to the accumulation of toxic and immunogenic CD-related epitopes,  (Only according to results of association analysis are not sufficient for candidate gene identification.)

and iii) develop useful molecular markers to select durum wheat varieties with reduced influence on the immunologic mechanisms of digestive disorders. (significant markers are only suggestive and have not been validated by any germplasm)

Therefore, this research provides very preliminary information, further research needs to conducted to draw clear conclusions.